# Comprehensive Study of Some Cyanobacteria in Moscow Waterbodies (Russia), Including Characteristics of the Toxigenic *Microcystis aeruginosa* Strains

**DOI:** 10.3390/toxins17100506

**Published:** 2025-10-14

**Authors:** Elena Kezlya, Elina Mironova, Ekaterina Chernova, Maria Gololobova, Andrei Mironov, Ekaterina Voyakina, Yevhen Maltsev, Dina Snarskaya, Maxim Kulikovskiy

**Affiliations:** 1K.A. Timiryazev Institute of Plant Physiology RAS (IPP RAS), 35 Botanicheskaya St., Moscow 127276, Russia; elechka-03@mail.ru (E.M.); diatomironov@yandex.ru (A.M.); ye.maltsev@gmail.com (Y.M.); 2Faculty of Biology, M.V. Lomonosov Moscow State University, Leninskie Gory 1, Building 12, Moscow 119234, Russia; gololobovama@mail.ru; 3Scientific Research Centre for Ecological Safety, St. Petersburg Federal Research Center, Russian Academy of Sciences, St. Petersburg 197110, Russia; s3561389@yandex.ru (E.C.); katerina.voyakina@gmail.com (E.V.); 4SPBU Research Park, Centre for Culture Collection of Microorganisms, Botanicheskaya St., 17A, Peterhof, St. Petersburg 198504, Russia; dina.snarskaya@spbu.ru; 5All-Russian Collection of Microorganisms (VKM), Scryabin Institute of Biochemistry and Physiology of Microorganisms, Russian Academy of Sciences, Pushchino 142290, Russia; max-kulikovsky@yandex.ru

**Keywords:** cyanotoxin screening, microcystin, 16S rRNA phylogeny, HPLC-HRMS, polyphasic approach

## Abstract

The current study focuses on the diversity, distribution and toxic potential of cyanobacteria in the waterbodies of Moscow, Russia. The research involves the sampling of natural and artificial water environments situated within the Moscow city agglomeration, including the waterbodies of recreational importance. A total of 20 strains of cyanobacteria, namely representatives of *Anabaena*, *Aphanizomenon*, *Argonema*, *Dolichospermum*, *Microcystis* and *Woronichinia*, are isolated from the collected samples. The morphology of the newly obtained strains is analyzed through light microscopy. The results of morphological identification are compared to the molecular data. The molecular phylogeny of the cyanobacterial strains is assessed on the basis of 16S rRNA sequencing. The detection of cyanotoxin-producing genes through PCR reveals two strains of *Microcystis aeruginosa* capable of microcystin synthesis. Further analysis using HPLC-HRMS demonstrates that microcystin production includes a high proportion (20–28%) of exceptionally toxic microcystin–leucine arginine compounds. Hereby, we discuss the morphology and phylogeny of the analyzed strains and provide comments on the toxic potential of cyanobacteria within the waterbodies of Moscow.

## 1. Introduction

Cyanobacteria are widespread photoautotrophic prokaryotes occurring in fresh, brackish and marine ecosystems and inhabiting both planktonic and benthos communities. Under certain conditions, different species form massive blooms and demonstrate the ability to produce toxins. The action of hepatotoxins, neurotoxins and dermatotoxins is known among cyanotoxins, as well as toxins that inhibit protein synthesis. Cyanobacterial toxins are divided into three main groups based on the chemical structure: cyclic peptides, alkaloids and lipopolysaccharides [1]. Blooms of toxic cyanobacteria impact health and ecosystem viability worldwide. In recent years, the frequency and intensity of CyanoHABs have risen due to the development of anthropogenic activities and global climate change [2,3,4].

The study of cyanobacteria that cause CyanoHABs is currently extremely relevant. This is confirmed by the numerous publications devoted to this problem [3,5,6,7]. The data on cyanobacterial species composition and distribution, especially toxic and potentially toxic ones, are especially important under megalopolis conditions for biosafety control and human health protection (particularly for recreational waterbodies) [3]. Currently, for the detection and identification of cyanobacteria, a polyphasic approach, including the tools of molecular biology as well as genes related to the producing cyanotoxins, is used [6,8].

In Russia, the genetic diversity and toxicogenic potential of cyanobacteria are poorly studied. Studies of cyanotoxins in different species, using analyses of cyanotoxin-encoding genes in environmental samples, demonstrate their ability to produce toxins. Chemical analysis through liquid chromatography coupled with mass spectrometry has been started recently [9,10,11,12,13,14,15,16]. However, studies of cyanobacteria using a polyphasic approach to evaluate the toxic potential in Russia are still limited. Although there is evidence of the presence of cyanotoxins in different waterbodies of Russia, their producers are often not determined.

The current study focuses on the waterbodies of Moscow, the largest city of Russia. The Moscow hydrological system comprises about 200 rivers and streams and more than 600 ponds. Among them, the Moskva River is the main water artery of the city [17]. However, the data on cyanobacteria and cyanotoxin occurrence in this watercourse, as well as other waterbodies in Moscow, are lacking. There are some studies that present the results of the quantitative monitoring of cyanobacteria in water sources of Moscow and assess the effectiveness of water purification to expel cyanobacteria [18,19]. Several studies are devoted to assessing the water quality of the Moskva River based on the analysis of the dynamics of algal communities, including cyanobacteria [20,21]. It is worth mentioning that some studies on Moscow’s waterbodies investigate the problems of blooms of potentially toxic species of cyanobacteria, e.g., blooms of *Microcystis aeruginosa* (Kützing) Kützing in Chystiy Pond [22]. However, there are no studies using a polyphasic approach to characterize cyanobacteria in the waterbodies of Moscow as of yet.

In this regard, the aim of our study is to isolate monoclonal cyanobacterial strains from different waterbodies of Moscow, to screen potentially toxic cyanobacteria for the ability to produce toxins, to characterize the detected toxic species based on morphology and molecular genetic data and to determine the profile and quantitative content of structural groups of the cyanotoxins revealed.

## 2. Results

As a result of the study, 20 strains of cyanobacteria (class Cyanophyceae A.W.Bennett & G.Murray) were obtained from 12 natural samples of phytoplankton and benthos: from order Nostocales Borzi, family Aphanizomenonaceae Elenkin (*Aphanizomenon* Morren ex Bornet & Flahault sp., *Dolichospermum* (Bornet & Flahault) P.Wacklin, L.Hoffmann & Komárek sp., *Anabaena* Bory ex Bornet & Flahault sp., one strain), from order Oscillatoriales Schaffner, family Phormidiaceae Anagnostidis & J.Komárek (*Argonema galeatum* Skoupý & Dvořák, one strain) and from order Chroococcales Schaffner, family Microcystaceae Elenkin (*Microcystis aeruginosa*, two strains and *Woronichinia naegeliana* (Unger) Elenkin, 14 strains).

### 2.1. Morphological Analysis


***Dolichospermum* sp. strain CBMC469m (Figure 1A)**


Trichomes free-floating, solitary, straight or slightly irregularly curved, clearly constricted at the cross-walls, without mucilaginous envelope. Cells spherical to slightly elliptical, sometimes barrel-shaped, olive-green, (8.0) 9.0–12.0 × 9.0–10.5 μm, end cells undifferentiated from other vegetative cells. Heterocytes spherical, solitary, intercalary, 10.5–16.0 μm in diameter. Akinetes were not found in culture. The strain CBMC469m was isolated from a plankton sample M4 (Moskva River, Krylatskoye district, see Section 5, “Materials and Methods”).


***Aphanizomenon* sp. strain CBMC479m (Figure 1B)**


Trichomes solitary, straight or slightly curved. Cells cylindrical, barrel-shaped to almost spherical, 4.0–6.5 × 3.0–4.0 μm, towards the ends cylindrical, elongated with many aerotopes, olive-green; terminal cells usually more elongated (sometimes slightly narrower) 6.0–9.0 × 2.8–3.7 μm, hyaline, with aerotopes. Heterocytes intercalary, usually one, rarely two per trichome, cylindrical, 6.2–6.5 × 4.0 μm. Akinetes were not found in culture. The strain CBMC479m was isolated from a plankton sample M10 (Meshchersky Pond, see Section 5, “Materials and Methods”).


***Anabaena* sp. strain CBMC473m (Figure 1C)**


Trichomes straight or flexuous, without mucilaginous envelopes, blue-green, more or less cylindrical, distinctly constricted at the cross-walls. Cells barrel-shaped, ± isodiametric or slightly longer or shorter than wide, 5.0–7.5 × 5.2–5.5 μm, blue-green, terminal cells rounded or conical. Heterocytes solitary, intercalary, spherical to cylindrical or elliptical, 6.5–11.0 × (6.0) 6.5–7.6 μm from 2 to 8–12 per trichome. Akinetes were not found in culture. The strain CBMC473m was isolated from a benthic sample M7 (Nizhny Fermsky Pond, see Section 5, “Materials and Methods”).


***Argonema galeatum* strain CBMC475m (Figure 2)**


Filaments wavy, sometimes spirally twisted together, blue-green to gray-green in color. The cells are 8.0–9.2 µm wide and 1.2–2.8 (3) µm long. The sheaths colorless to light brown, thin, distinct. The sheath can exceed filament or not. The true branching was not observed. Trichomes cylindrical, not attenuated with a concave apical cell, slightly or not constricted at the cross-walls. Necridic cells present. The strain was isolated from a benthic sample M9 (unnamed pond in Odintsovsky district, see Section 5, “Materials and Methods”).


***Woronichinia naegeliana* (Figure 3, Figure 4 and Figure 5)**


All studied strains had similar morphological characteristics. Colonies microscopic, spherical, ellipsoid or spherical–irregular with radially arranged cells in the peripheral monolayer, attached at the ends with gelatinous tubular stalks. Colonies kept form during culturing (checked after 6 months of cultivation). Mucilaginous envelope around the colony colorless. Cells ovoid or ellipsoid, 3.0–5.2 µm wide and 5.0–8.0 µm long, blue-green or olive-green with numerous aerotopes.

*Woronichinia naegeliana* was found as the dominant or co-dominant species in plankton of an unnamed pond in the Odintsovsky district (sample site M9, strains CBMC677m, CBMC678m, CBMC680m, CBMC681m, CBMC683m, CBMC682m), in plankton of Meshchersky Pond (sample site M10, strains CBMC687m, CBMC685m, CBMC686m, CBMC687m, CBMC689m, CBMC690m, CBMC691m), in plankton of Nizhny Fermsky Pond (sample site M7, strains CBMC672m, CBMC673m, CBMC674m, CBMC675m) and in plankton of the Moskva River in the Krylatskoye district (sample site M4, strain CBMC671m). For details about the sampling sites, see Section 5, “Materials and Methods”.


***Microcystis aeruginosa* strains CBMC403m, CBMC523m (Figure 6)**


Colonies microscopic, irregular in outline, loose, diffluent in the culture in both strains. Mucilage colorless, structureless, sometimes forming a distinct margin around cells. Cells spherical or elongate before division, olive-green, (3.0) 3.8–6.0 µm in diameter with numerous aerotopes. Aerotopes can disappear in old culture and when storing the culture in the refrigerator at 10 °C.

*Microcystis aeruginosa* was found as being dominant in the plankton of Meshchersky Pond (sampling site M10, strain CBMC523m) and in the plankton of an unnamed pond in Marfino district No1 (sampling site M13, strain CBMC403m). For details about the sampling sites, see Section 5, “Materials and Methods”.

### 2.2. Molecular Analysis

Phylogenetic analysis was based on 16S rDNA sequencing performed for the newly obtained strains of cyanobacteria from waterbodies of the Moscow region. The selected reference sequences from GenBank were also included in the analysis. Using the BI and RAxML analyses, we constructed two separate trees—the first (Figure 7), to assess the phylogeny of the *Microcystis* Lemmermann-*Woronichinia* Elenkin species complex—the second (Figure 8), for the representatives of Nostocales and Oscillatoriales analyzed in the current study.

The first phylogram (Figure 7) focuses on the genus *Microcystis* and its closest related genus, *Woronichinia*. Our phylogenetic analysis demonstrates a close relationship between *Microcystis* strains. They form a highly supported clade (BI = 100, ML = 1). The vast majority of reference strains cluster with no support from BI and RAxML. The control strain *Microcystis aeruginosa* CALU972 and the *M. aeruginosa* CBMC403m and CBMC523m strains studied here were positioned near other *Microcystis* strains. The results of a pairwise similarity analysis based on *p*-distance showed that the sequences of the 16S rRNA gene of the studied strains of *M. aeruginosa* (CBMC403m and CBMC523m) and type strains (NIES-1082, NIES-1123, NIES 104) of the species and some others from our dataset (n = 16) are completely identical or have a high similarity (>99.8%) (see Appendix A). All 12 newly obtained strains of *Woronichinia naegeliana* form a highly supported subclade (BI = 99, ML = 1), while strains *W. naegeliana* and *Woronichinia* sp. from GenBank near the subclade included *Snowella* Elenkin species. It is also worth mentioning that the sister relationships between *Woronichinia* and *Snowella* are well-supported (BI = 88, PP = 1). The results of a pairwise similarity analysis of the studied *Woronichinia* strains showed a complete or high similarity (>99.1%) with type strain *W. naegeliana* 0LE35S01 (Appendix A).

The second phylogram (Figure 8) was aimed at reconstructing the phylogeny of the representatives of orders Nostocales and Oscillatoriales from our collection. The dataset for this molecular tree was supplemented with strains of *Dolichospermum*, *Aphanizomenon*, *Anabaena*, *Trichormus* (Ralfs ex Bornet & Flahault) Komárek & Anagnostidis, *Argonema i* Skoupý & Dvorák, *Oscillatoria* Vaucher ex Gomont, *Microcoleus* Desmazières ex Gomont, *Phormidium* Kützing ex Gomont, *Planktothrix* Anagnostidis & Komárek and *Cyanobium* Rippka & Cohen-Bazire from GenBank. Strain *Dolichospermum* sp. CBMC469m is hereby positioned within the major clade of *Dolichospermum*. The clade is highly supported by BI analysis (PP = 1). A single strain of *Dolichospermum* cf. *lemmermannii* (Richter) Wacklin, Hoffmann & Komárek (strain ID “262”) clusters separately with clade *Aphanizomenon* strains. *Aphanizomenon* strains form a separate clade with a high BI support (PP = 0.98). A single strain of ours—*Aphanizomenon* sp. CBMC479m—places near the strain *Aphanizomenon gracile* Lemmermann NIVA-CYA851 in the *Aphanizomenon* clade, whereas the studied strain *Anabaena sp.* CBMC473m is not affiliated with the clade *Anabaena*/*Trichormus*, but forms an independent line with maximum support from LB = 100, PP = 1. Strains of *Argonema* and *Oscillatoria* form a separate clade with significant support from LB = 96, PP = 1. Our strain *Argonema galeatum* CBMC475m occupies a position with other strains of this species in the *Argonema galeatum* subclade. The comparison of the 16S rRNA sequences revealed a 100% similarity between type strains of *A. galeatum* and the studied strain CBMC475m (Appendix A).

### 2.3. Detection of Cyanotoxin-Producing Genes by PCR

A PCR amplification of the genes involved in microcystin (*mcy*ACa, *mcy*B and *mcy*E), cylindrospermopsin (*cyr*B) and anatoxin (*ana*C) production was carried out for all 20 strains. As a result, only two strains of *Microcystis aeruginosa* (CBMC403m and CBMC523m) had fragments of MC-encoding genes (*mcy*ACa, *mcy*B and *mcy*E) (Appendix A). Accordingly, for these two strains and the control strain CALU972, the content of microcystins was determined using HPLC-HRMS.

### 2.4. Detection of Microcystins by HPLC-HRMS

The identified microcystin congeners and their concentrations are presented in Table 1. Identification was based on the exact masses of MCs and the data from the fragment spectra. The fragment spectra were obtained for the identified compounds and standards corresponding to each other (Appendix A).

All of the identified compounds belong to the microcystin group because of the presence of some *m*/*z* corresponding to the characteristic fragments of microcystin moleculars, such as 135.08 (Adda-fragment [C_9_H_11_O]^+^), *m*/*z* 213.09 ([Glu-Mdha+H]^+^) and *m*/*z* 375.19 ([C_11_H_15_O+Glu+Mdha]^+^) in the fragment spectra. Other characteristic *m*/*z* values for each structural variant of MC and the corresponding structure fragments are listed in Table 2.

In the biomass samples, seven structural variants of microcystins differing in toxicity were identified. All detected structural variants were arginine-containing. The contribution of the most toxic representative, MC-LR, to the total microcystin content was 20–28% for strains CBMC403m and CBMC523m, and was completely absent in the CALU972 sample. The maximum contribution to the microcystin composition in strains CBMC403m and CBMC523m was made by MC-YR. For the microcystin profile of strain CALU972, the most characteristic were demethylated variants of MC-RR ([D-Asp^3^, Dhb^7^]MC-RR; [D-Asp^3^]MC-RR) and a demethylated variant of MC-LR ([D-Asp^3^]MC-LR). The extracted ion chromatogram of high resolution for the MC congeners detected in the biomass of strains CBMC403m, CALU972 and CBMC523m is presented in Figure 9.

## 3. Discussion

In this study, cyanobacteria from waterbodies in Moscow were studied for the first time with the help of a polyphasic approach. Out of the 20 strains obtained, potentially toxic taxa [23,24] included representatives of the family Aphanizomenonaceae (Nostocales)—*Aphanizomenon* sp., *Dolichospermum* sp. and *Anabaena* sp.—and representatives of Microcystaceae (Chroococcales)—*Microcystis aeruginosa*. However, data on the toxic potential of the representatives of *Argonema* and *Woronichinia* were limited. As a result of the screening of strains with PCR amplification of the MC- (*mcy*A-C, *mcy*B, *mcy*E), CYN- (*cyr*B) and ANA- (*ana*C) encoding genes, we were able to detect MC-encoding genes in two strains of *Microcystis* sp. (CBMC403m and CBMC523m), in addition to the control strain *Microcystis aeruginosa* CALU972. It should be noted that the PCR screening method does not prove 100% the presence of the corresponding gene clusters, since NRPS/PKS gene clusters are modular, and related modules may appear in different operons, coding the synthesis of completely different products.

A further analysis of the strains CBMC403m and CBMC523m through HPLC-HRMS confirmed the content of microcystins in the biomass. Microcystins are hepatotoxins exhibiting toxicological effects on multiple organs, but accumulating primarily in the liver, thus causing hepatotoxicity. In the studied samples, seven structural variants were found, of which a significant proportion (20–28%) in strains CBMC403m and CBMC523m is microcystin–leucine arginine (MC-LR), one of the most common and toxic compounds in the MC family [25]. MC-LR shows neurotoxicity, nephrotoxicity, cardiotoxicity and reproductive toxicity. It is worth noting that the toxigenic *Microcystis aeruginosa* strain CALU972, used as a control hereby, has been stored in the collection for more than 30 years, and its toxicity has been repeatedly confirmed [26,27]. In this study, we also confirm the toxicity of this culture on the basis of PCR and HPLC-HRMS.

There are several known ways that humans can be exposed to microcystins. In particular, affection can occur as a result of contaminated water ingestion, the consumption of fish from lakes and reservoirs that routinely experience significant toxic *Microcystis* blooms or body contact during swimming in contaminated waters (causing rashes, itching and other allergic responses in the skin, eyes, ears or respiratory organs) [25]. Notably, toxic strain CBMC523m of *M. aeruginosa* was isolated from a recreational reservoir Meshchersky Pond (sampling site M10), which is actively used for swimming and fishing during the summer. Based on morphology, we registered the massive development of *Microcystis aeruginosa* specimens in the plankton of this reservoir, as well as in the plankton of an unnamed pond in Marfino district #1 (sampling site M13), from which the second toxigenic strain of *M. aeruginosa* CBMC403m was isolated. Notably, cyanobacteria of the genus *Microcystis* are known to be among the most common producers of cyanotoxins (i.e., microcystins) in fresh waterbodies on all continents except Antarctica. According to the review by Harke et al. [28], *Microcystis* blooms were recorded in 108 countries, of which the presence of cyanotoxins was detected at 79 localities. At the same time, the ability to produce certain cyanotoxins is species-specific, toxin-specific and geographically diverse [29], while the factors regulating the development of toxin-producing and non-toxin-producing cyanobacteria of the same species remain unclear [30]. Hitherto, metagenome analysis using high-throughput sequencing approaches is an effective tool for determining the distribution and development of species in the presence of detailed characterized nucleotide sequences. This is of particular importance in terms of the study of toxigenic species.

This underlines the relevance of the present study, as well as the need to study the spread of toxic cyanobacteria at least in recreational reservoirs of Moscow and in reservoirs used for the central water supply in the city. The toxic cyanobacteria found indicate the importance of regular monitoring to assess the development of toxic microalgae under conditions of anthropogenic pressure on waterbodies in the city and global climate change. As far as we know, the targeted monitoring of cyanobacterial blooms and assessment of cyanotoxin content in surface reservoirs is not currently performed in Moscow. The obtained reference nucleotide sequences (molecular barcodes) of these strains will further make it possible to accurately assess their distribution and degree of development in reservoirs and watercourses of Moscow as a part of further metagenomic studies.

In addition, we investigated the strains of *Argonema galeatum* (Oscillatoriales, Phormidiaceae) and *Woronichinia naegeliana* (Chroococcales, Microcystaceae) in detail. Regarding *Argonema*, as of yet, little data have accumulated on the toxic potential of the representatives of this genus. The genus itself has been described recently by Skoupý et al. [31] from soil crusts of James Ross Island (Western Antarctica). However, the authors investigated the distribution of *Argonema* based on a search of amplicons in the NCBI SRA [32] and the NCBI nucleotide database and discovered 57 possible *Argonema* matches at 57 geographical localities in all the world. The toxic potential of *Argonema* was not determined in that study. Hereby, we provide the first record of *A. galeatum* in Russia. The record is supported by both morphological (Figure 4) and molecular (Figure 9) analyses. MC-, CYN- and ANA-encoding genes were not detected, which is why the strain *A. galeatum* CBMC475m is characterized as non-toxigenic.

*Woronichinia naegeliana* is known to cause massive blooms in waterbodies worldwide [33,34,35,36], including Russia [37]. However, the first reported isolates of the genera were characterized by 16S rRNA gene sequencing and morphological analysis back in 2006 [38]. *mcy*E-specific PCR demonstrated that the strains did not produce microcystins. Later, Bober et al. [36] reported the trace amounts of microcystin-LR extracted from *W. naegeliana*, as well as three types of oligopeptides: microginins, cyanopeptolins and anabaenopeptins. In the latter research, Bober and Bialczyk [35] linked the toxic effect of *W. naegeliana* to the invertebrate zooplankton with microginin.

Recently, Dreher et al. [39] determined the complete genome of *Woronichinia naegeliana* WA131 from a US Pacific NW freshwater lake and confirmed that the genome contains no genes responsible for cyanotoxin biosynthesis. At the same time, the authors noted the presence of gene clusters of anabaenopeptins, cyanopeptolins, microginins and some post-translationally modified peptides. As a result, *W. naegeliana* was characterized as non-toxigenic. In this study, strains of this species were obtained from waterbodies of Moscow for the first time. The detected strains caused biomass blooms in four water reservoirs. The analysis of the morphology and sequences of gene 16S rRNA clearly confirms the results of a primary identification. A PCR screening of the strains regarding the presence of MC-, CYN- and ANA-encoding genes did not yield positive results, which, again, indicates the non-toxigenicity of *W. naegeliana*.

## 4. Conclusions

In this study, an investigation of cyanobacteria from the waterbodies of Moscow using a polyphasic approach was conducted for the first time. The morphology, phylogeny and toxic potential of 20 strains were studied in detail. As a result, two toxigenic strains of *Microcystis aeruginosa* (CBMC403m and CBMC523) were found to produce microcystin hepatotoxins with a high proportion (20–28%) of MC-LR, one of the most toxic compounds in the MC family. One of the analyzed strains (CBMC523m) was isolated from a recreational reservoir—Meshchersky Pond—that is actively used for swimming and fishing, which indicates the importance of regular monitoring to assess the development of toxic cyanobacteria. In general, our research reveals that the coordination of molecular, morphological and environmental data on cyanobacteria and the assessment of their toxigenic potential is a priority and an urgent task of modern science. Monoclonal cultures allow us to characterize the genetic and physiological characteristics of specific strains, and finally confirm their ability to synthesize toxins. The replenishment of databases with well-documented nucleotide sequences is the basis for accurate identification, the study of phylogeny and biogeography of taxa, the identification of invasions and the decoding of metabarcoding data. Newly acquired data about the distribution of toxigenic taxa of cyanobacteria could serve as the basis for the development of effective strategies for monitoring harmful algal blooms and controlling the quality and safety of natural waters in Moscow.

## 5. Materials and Methods

### 5.1. Sample Collection and Preparation

Field study was conducted in June–July 2024 across various locations in Moscow and the Moscow region. Sampling was performed at 12 aquatic sites, i.e., the samples were collected in the Moskva River, Krylatskoye Rowing Canal and 10 ponds situated in Krylatskoye, Marfino, Odintsovsky and Ostankinsky districts of Moscow city. The sampling sites were chosen due to their recreational role and exposure to significant anthropogenic pressure. At each sampling site, phytoplankton and benthic samples were collected.

For samples of phytoplankton, Apstein nets (with mesh size = 29 µm) were utilized; 100 liters of water were passed through the net with a bailer, and the collected concentrate (250 mL) was transferred to a test tube. Benthic samples were collected with brushes from silty stones, biofilms and sand. Duplicates of benthic samples from each location were fixed with a 40% formaldehyde solution to a final concentration of 4%.

Hydrochemical parameters were measured at the time of sample collection using the Hanna Combo (HI 98129) handheld device (Hanna Instruments, Inc., Woonsocket, RI, USA). Information about the collected samples and corresponding hydrochemical measurements is summarized in Table 3. Hydrological parameters of the sampled waterbodies are presented in accordance with the “Temnyj Les” website [40].

### 5.2. Culturing, Microscopy and Morphological Identification

After collection, samples were transferred to Petri dishes (d = 60 mm) and filled with WC liquid medium [41]. For two weeks, single colonies of *Microcystis* and *Woronichinia* and filaments were isolated under a Zeiss Axio Vert A1 inverted light microscope (ZEISS, Oberkochen, Germany). They were washed three–five times and placed in wells on microtiter plates with 300 µL WC liquid medium. Afterwards, the cultures were incubated at 20–22 °C under a light intensity of 4000 lux with a 12:12 h light:dark photoperiod. The strains were placed into Petry dishes (diameter 40 mm) with WC liquid medium after two–three weeks. Original strains were deposited in the Culture and Barcode Collection of Microalgae and Cyanobacteria “Algabank” (CBMC) in K. A. Timiryazev Institute of Plant Physiology RAS.

Morphological examination was conducted using a Zeiss Axio Scope A1 microscope equipped with a stock oil immersion objective (×100, n.a. 1.4, DIC) and a Zeiss Axiocam ERc 5s camera (ZEISS, Oberkochen, Germany). Morphological identification was carried out based on LM observations, using standard taxonomic keys for cyanobacteria [42,43].

### 5.3. DNA Extraction and Amplification

After cultivation, biomass of strains was centrifuged in 1.5 mL eppendorfs for 5 min at 6000 rpm. DNA was extracted by Chelex100 Chelating Resin (protocol 2.2, Bio-Rad Laboratories, Hercules, CA, USA). PCRs were conducted with ScreenMix (Evrogen, Moscow, Russia). Amplification of the 16S rRNA gene was performed with CYA359F [44] and 1467R [45] primers. Amplification conditions for the 16S rRNA gene were as follows: initial denaturation for 5 min at 95 °C followed by 35 cycles of 30 s denaturation at 94 °C, 30 s annealing at 52 °C and 50 s extension at 72 °C, with the final extension for 10 min at 72 °C.

Genes *mcy*E/*nda, mcyA*, *mcyB* and *mcy*E were targeted for MC production potential; the anatoxin-a gene (*ana*C) and cylindrospermopsin gene target region (*cyr*B) were amplified using specific primer pairs available in the literature (Table 4).

Amplification conditions were as follows: initial denaturation for 5 min at 94 °C followed by 30 cycles of 30 s denaturation at 94 °C, 30 s annealing from 52 to 59 °C, according to Table 4, and 60 s extension at 72 °C, with the final extension for 10 min at 72 °C.

The products of amplification were visualized through horizontal electrophoresis in agarose gel (1.0%) stained with SYBR™ Safe stain (Life Technologies, Carlsbad, CA, USA). Obtained products were purified using a premade mixture of FastAP, 10× FastAP Buffer, Exonuclease I (Thermo Fisher Scientific, Waltham, MA, USA) and sterile water. Fragments of the 16S rRNA gene were sequenced on both strands using forward and reverse PCR primers and the BigDye^®^ Direct Cycle Sequencing (Applied Biosystems, Waltham, MA, USA), followed by electrophoresis using a Genetic Analyzer 3500 sequencer (Applied Biosystems, Waltham, MA, USA).

### 5.4. HPLC-HRMS Toxin Analysis

For HPLC-HRMS toxin analysis, the biomass of strains was transferred to Erlenmeyer flasks (400 mL) and filled with WC liquid medium up to 200 mL. Further cultivation was performed for 4–6 weeks at constant conditions (20–22 °C, 4000 lux, 12:12 h light:dark photoperiod) until reaching a stationary growth phase. After cultivation, the grown-up biomass was washed with distilled water (for three times) and sedimented in 50 mL tubes. The sedimentation was conducted in a centrifuge for 6 min at 12000 rpm. Sedimented biomass was preserved at −80 °C and freeze-dried in a Labconco FreeZone 2.5 L Benchtop Freeze Dryer (Labconco Corporation, Fort Scott, KS, USA).

To detect cyanotoxins, analytical-grade chemicals were used. Acetonitrile (HPLC-grade) and methanol (LiChrosolv hypergrade for LC-MS) were purchased from Merck (Darmstadt, Germany); formic acid (98–100%) was obtained from Fluka Chemika (Buchs, Switzerland). High quality water (18.2 MΩ cm^−1^) was produced by the Millipore Direct-Q water purification system (Bedford, MA, USA). The MC-LR, MC-YR and MC-RR standards were purchased from Sigma Aldrich; the MC-LA, MC-LF, MC-LY, MC-LW, [D-Asp^3^]MC-LR and [D-Asp^3^]MC-RR were from Enzo Life Sciences, Inc., New York, USA.

To define the profile of toxins and quantify them, the high-performance liquid chromatography–high-resolution mass-spectrometry (HPLC-HRMS) method was used.

The sample preparation procedures involved the extraction of cyanotoxins from liophilizated biomass using treatment with 75% methanol in an ultrasonic bath (FinnSonic OY M03, Finland; US frequency 40 kHz, US power 80W/160W (nominal/peak); treatment time (extraction)—20 min).

Analyses of extracts were performed using the LC-20 Prominence HPLC system (Shimadzu, Japan) coupled with a Hybrid Ion Trap-Orbitrap Mass Spectrometer—LTQ Orbitrap XL (Thermo Fisher Scientific, San Jose, CA, USA) according to Chernova et al. [52]. Separation was achieved by gradient elution using a Thermo Hypersil Gold RP C18 column (100 × 3 mm, 3 μm) with the mobile phase consisting of water and acetonitrile both containing 0.1% of formic acid.

Mass spectrometric analysis was carried out using electrospray ionization in the positive ion detection mode. The identification of target compounds was based on the accurate mass measurement of [M+H]^+^ or [M+2H]^2+^ ions (resolution of 30,000, accuracy within 5 ppm), the collected fragmentation spectrum of the ions and the retention times. The concentrations of the toxins were calculated based on the peak area of standards analyzed on the same day and under the same conditions. For some identified MC variants, we did not have standard compounds; their quantification was performed as follows: [D-Asp^3^, Dhb^7^]MC-RR and [D-Asp^3^]MC-YR were quantified using the information of the signal area corresponding to the [D-Asp^3^]MC-RR (as signal for double charged molecular) and MC-LR, accordingly. Procedural LOD and LOQ from freeze-dried biomass were 0.3 μg g^−1^ and 0.9 μg g^−1^.

### 5.5. Phylogenetic Analysis

The analysis of phylogeny was conducted to assess the phylogeny of *Woronichinia* and *Mycrocystis* (Figure 7) and other groups of cyanobacteria (Figure 8). The procedures of phylogenetic analysis were identical for the two phylograms.

Newly obtained sequences were manually edited in Ridom TraceEdit (Ridom© GmbH, Münster, Germany). The sequences from forward and reverse primers were assembled pairwise using MEGA11.5 software (Pennsylvania State University, Pennsylvania, PA, USA [53]). The datasets were aligned with G-INS-I algorithm with Mafft ver. 7 software (RIMD, Osaka, Japan [54]). Unpaired regions were erased, and the aligned sequences were united into concatenated datasets. The resulting dataset for the *Mycrocystis*-*Woronichinia* phylogram (Figure 7) included a total of 86 sequences (matrix length = 926 nd)—16 newly obtained sequences and 70 reference sequences from GenBank. The dataset for the second phylogram (Figure 8) included 87 sequences (matrix length = 906 nd)—4 new and 83 reference. All new sequences acquired in this study were deposited to GenBank. Accessions for all sequences, chromatograms of the 16S rRNA gene sequences and constructed datasets are presented in Appendix A.

Bayesian inference (BI) analysis was conducted in BEAST ver. 1.10.1 software (BEAST Developers, Auckland, New Zealand [55]) with the following parameters: Yule process tree prior speciation model, GTR+G+I substitution model, 10 MCMC analyses employed for 10 million generations (burn-in 1000 million generations). The results of BI were processed in Tracer ver. 1.7.1 (MCMC Trace Analysis Tool, Edinburgh, UK [55]) and the initial 10% of trees were removed. Rapid bootstrapping and subsequent ML search (RAxML) of the concatenated alignment were performed in raxmlGUI 2.0 software [56]. RAxML was applied in order to assess the tree topology robustness with 1000 replicas, GTR substitution matrix and gamma substitution rates. The best scoring trees from BI and RAxML were viewed and manually edited in FigTree ver. 1.4.4 (University of Edinburgh, Edinburgh, UK) and Adobe Photoshop CC ver. 19.0 (Adobe, San Jose, CA, USA).

For *Argonema galeatum*, *Woronichinia naegeliana* and *Microcystis aeruginosa* the 16S rRNA sequences were also used to estimate the degree of similarity between gene sequences of type strains. Using MEGA11.5 software, the *p*-distances were determined to calculate the sequence similarity with the formula (1 − *p*) × 100. The results are present in Appendix A).

## Figures and Tables

**Figure 1 toxins-17-00506-f001:**
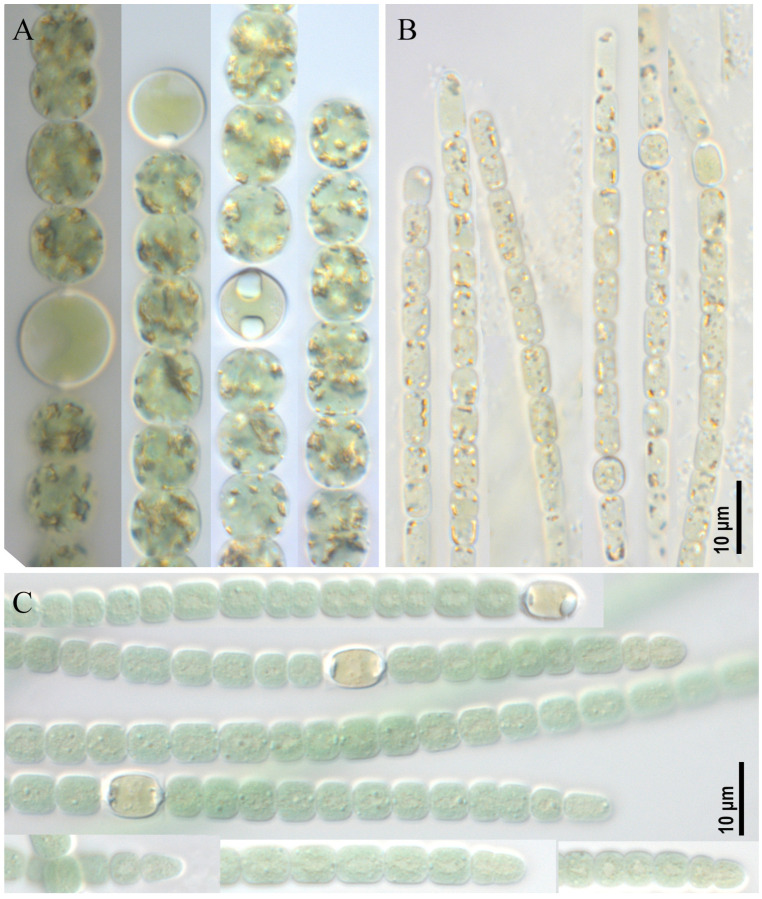
Studied strains: *Dolichospermum* sp. strain CBMC469m (**A**), *Aphanizomenon* sp. strain CBMC479m (**B**), *Anabaena* sp. strain CBMC473m (**C**).

**Figure 2 toxins-17-00506-f002:**
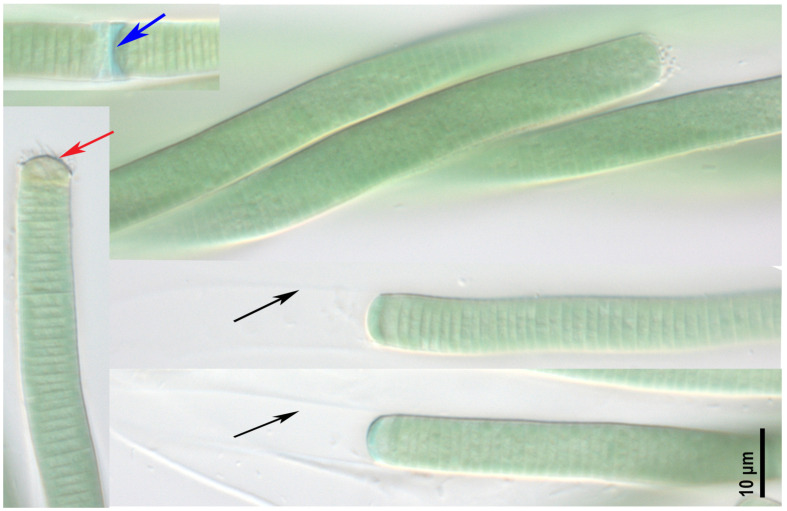
Studied strain *Argonema galeatum* CBMC475m. Blue arrow—necridic cell, red arrow—colored apical cell, black arrows—empty sheath.

**Figure 3 toxins-17-00506-f003:**
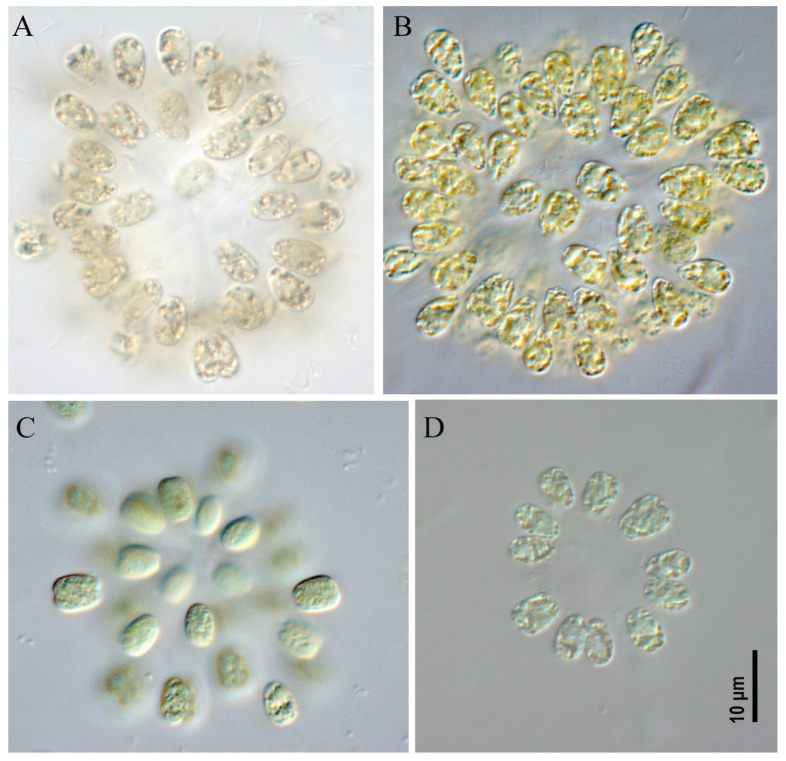
Studied strains *Woronichinia naegeliana*. CBMC678m (**A**), CBMC683m (**B**), CBMC687m (**C**), CBMC680m (**D**).

**Figure 4 toxins-17-00506-f004:**
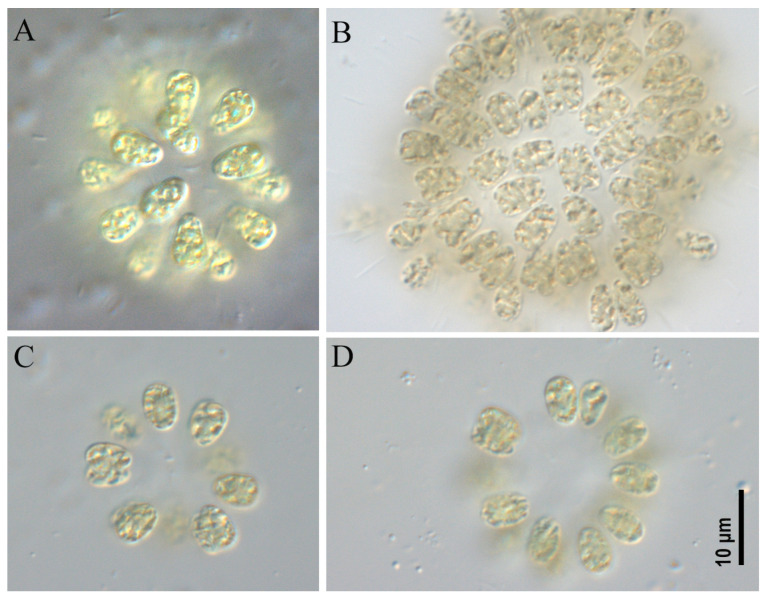
Studied strains *Woronichinia naegeliana*. CBMC674m (**A**), CBMC677m (**B**), CBMC689m (**C**), CBMC691m (**D**).

**Figure 5 toxins-17-00506-f005:**
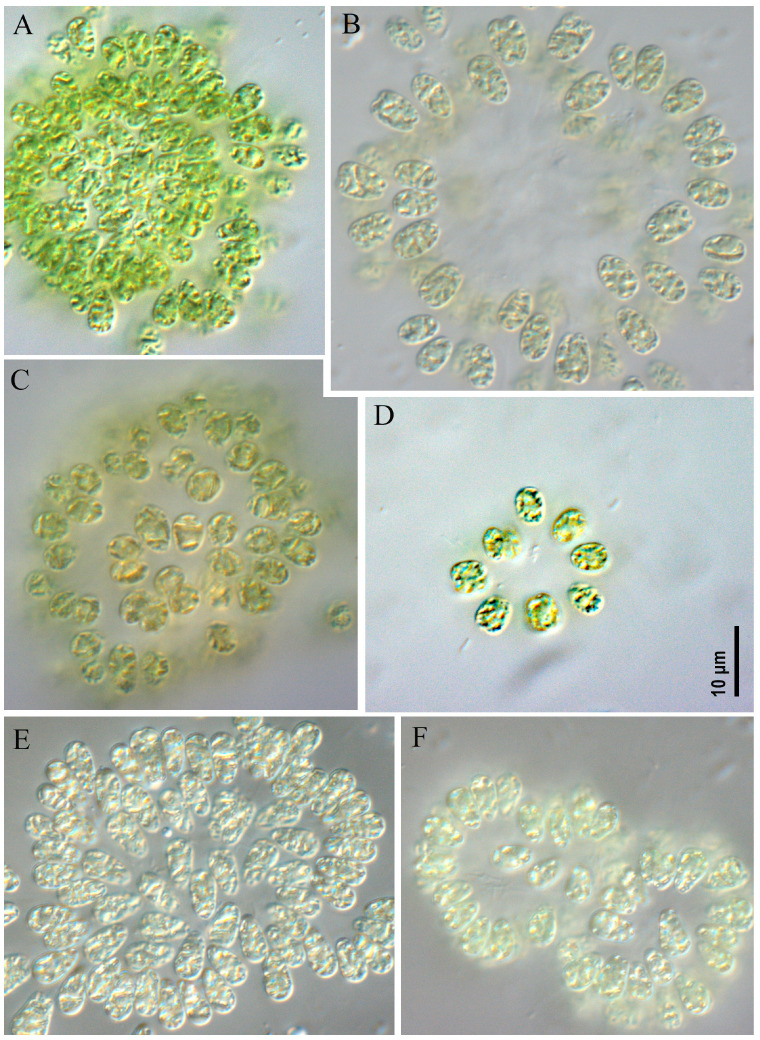
Studied strains *Woronichinia naegeliana*. CBMC685m (**A**), CBMC690m (**B**), CBMC681m (**C**), CBMC686m (**D**), CBMC672m (**E**), CBMC675m (**F**).

**Figure 6 toxins-17-00506-f006:**
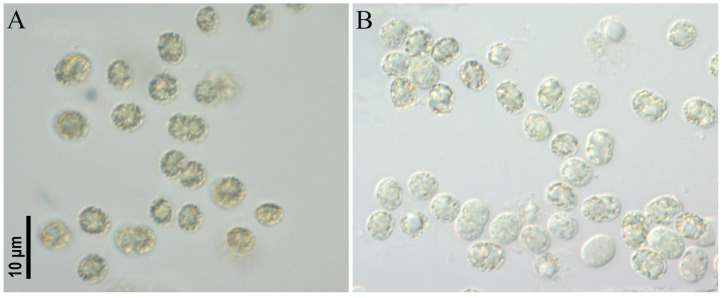
Studied strains *Microcystis aeruginosa* CBMC403m (**A**), CBMC523m (**B**).

**Figure 7 toxins-17-00506-f007:**
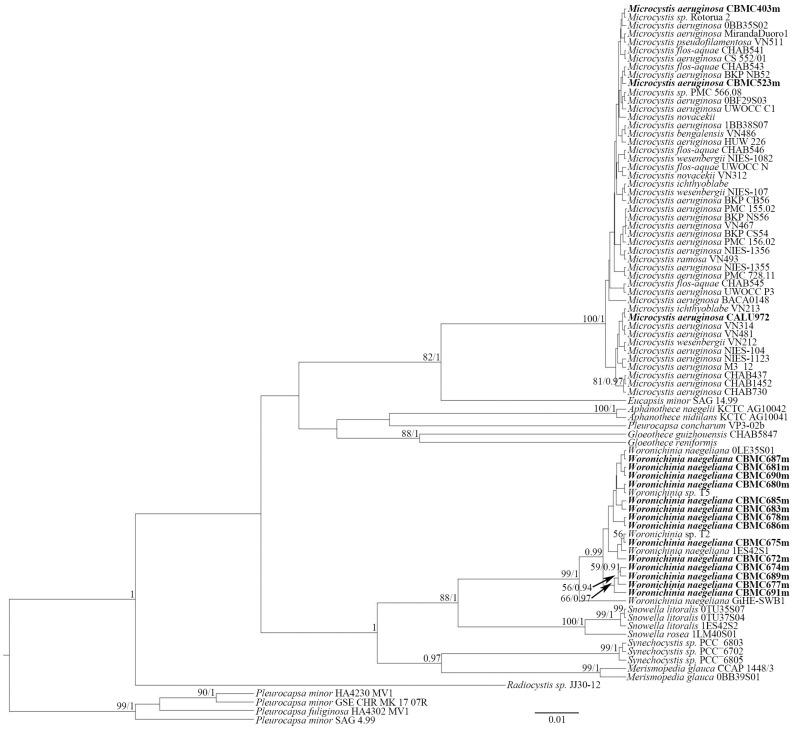
Phylogeny of *Microcystis*, *Woronichinia* and related genera based on BI and RAxML analyses (16S rRNA gene). Values of LB below 50 and PP below 0.90 are hidden. Strain numbers are indicated for all sequences. GenBank accessions and analysis procedure data are hidden (see Appendix A).

**Figure 8 toxins-17-00506-f008:**
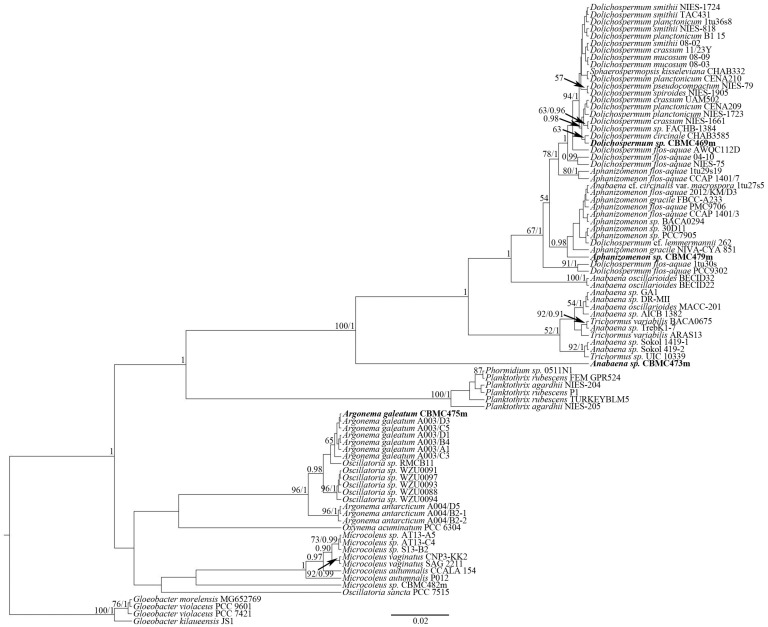
Phylogeny of selected cyanobacteria based on BI and RAxML analyses (16S rRNA). Values of LB below 50 and PP below 0.90 are hidden. Strain numbers are indicated for all sequences. GenBank accessions and analysis procedure data are hidden (see Appendix A).

**Figure 9 toxins-17-00506-f009:**
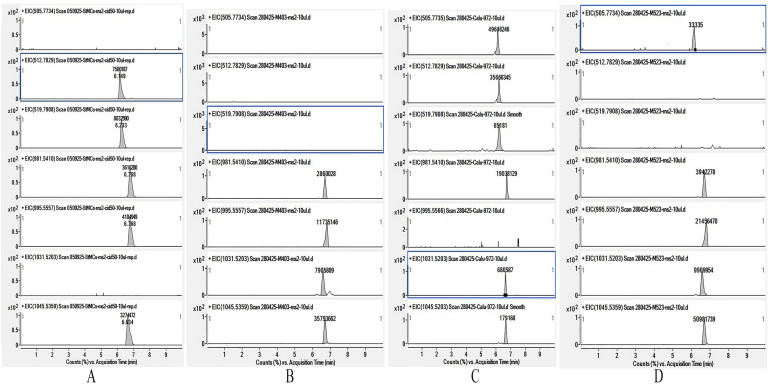
Extracted ion chromatogram of high resolution (mass accuracy within 5 ppm) for MC congeners (from top to bottom): *m*/*z* 505,7734 ([M+2H]^2+^ [D-Asp^3^, Dhb^7^]MC-RR); *m*/*z* 512.7829 ([M+2H]^2+^ [D-Asp^3^]MC-RR); *m*/*z* 519.79077 ([M+2H]^2+^ MC-RR); *m*/*z* 981.54095 ([M+H]^+^ [D-Asp^3^]MC-LR); *m*/*z* 995.55658 ([M+H]^+^ MC-LR); *m*/*z* 1031,5203 ([M+H]^+^ [D-Asp^3^]MC-YR); *m*/*z* 1045.53589 ([M+H]^+^ MC-YR) detected in (**A**) the solution of MC standards and biomass of strain (**B**) CBMC403m, (**C**) CALU972 and (**D**) CBMC523m, accordingly.

**Table 1 toxins-17-00506-t001:** Microcystin variants and their concentration (in μg/g freeze-dried biomass) in the studied strains.

Microcystin Congeners	Formula	*m*/*z*	CBMC403m	CALU972	CBMC523m
[D-Asp^3^,Dha^7^]MC-RR	C_47_H_71_N_13_O_12_	505.7751	<0.3	392.7	<0.9
[D-Asp^3^]MC-RR	C_48_H_73_N_13_O_1_	512.7829	<0.3	281.4	<0.3
MC-RR	C_49_H_75_N_13_O_12_	519.7908	<0.3	0.9	<0.3
[D-Asp^3^]MC-LR	C_48_H_72_N_10_O_12_	981.5410	23.5	147.9	24.2
MC-LR	C_49_H_74_N_10_O_12_	995.55658	96.6	<0.3	145.2
[D-Asp^3^]MC-YR	C_51_H_70_N_10_O_13_	1031.5203	65.5	118.3	3.4
MC-YR	C_52_H_72_N_10_O_13_	1045.5359	294.6	2.4	345.0
Total MCs			480.2	943.6	517.8

**Table 2 toxins-17-00506-t002:** Characteristic *m*/*z* values for each structural variant of MC and corresponding structure fragments.

MC Congener	Detected *m*/*z*	Charge of the Detected Ion	Characteristic Fragment Ions	Fragment Structure
MC-LR	995.5557	[M+H]^+^	446 553 599	[C_11_H_15_O-Glu-Mdha-Ala]^+^ [Mdha-Ala-Leu-MeAsp-Arg+H]^+^ [Arg-Adda-Glu+H]^+^
[D-Asp^3^]MC-LR	981.5410	[M+H]^+^	446 539 559 599	[C_11_H_15_O-Glu-Mdha-Ala]^+^ [Mdha-Ala-Leu-D-Asp-Arg+H]^+^ [C_11_H_15_O-Glu-Mdha-Ala-Leu]^+^ [Arg-Adda-Glu+H]^+^
MC-RR	519.7908	[M+2H]^2+^	440 453 596 599	[Mdha-Ala-Arg-MeAsp+H]^+^ [Arg-Adda+H−NH_3_]^+^ [Mdha-Ala-Arg-MeAsp-Arg+H]^+^ [Arg-Adda-Glu+H]^+^
[D-Asp^3^]MC-RR	512.7829	[M+2H]^2+^	426 499 582 599	[Mdha-Ala-Arg-D-Asp-H]^+^ or [Dha-Ala-Arg-MeAsp+H]+ [Ala-Arg-D-Asp-Arg+H]^+^ [Mdha-Ala-Arg-D-Asp-Arg+H]^+^ [Arg-Adda-Glu+H]^+^
[Dha,D-Asp^3^]MC-RR	505.7734	[M+2H]^2+^	412 568 599	[Dha-Ala-Arg-D-Asp-H]^+^ [Dha-Ala-Arg-D-Asp-Arg+H]^+^ [Arg-Adda-Glu+H]^+^
MC-YR	1045.5359	[M+H]^+^	446 599 603	[C_11_H_15_O-Glu-Mdha-Ala]^+^ [Arg-Adda-Glu+H]^+^ [Mdha-Ala-Tyr-MeAsp-Arg+H]^+^
[D-Asp^3^]MC-YR	1031.5203	[M+H]^+^	589 599 682 696	[Mdha-Ala-Tyr-D-Asp-Arg+H]^+^ [Arg-Adda-Glu+H]^+^ [Arg-Adda-Glu-Mdha+H]^+^ [DAsp-Arg-Adda-Glu-H_2_O+H]+

**Table 3 toxins-17-00506-t003:** List of collected samples with hydrological and hydrochemical parameters of water.

Sample Site ID	Name of Watercourse or Reservoir	Coordinates	Hydrological Parameters			Hydrochemical Parameters		
			Water Surface Area (km^2^)	Average Depth (m)	Recreational Use	Temperature (°C)	pH	Conductivity (mV)
M1	Bolshoy Krylatsky Pond	55.763134, 37.435047	0.125	2.5	Yes	23.4	8.09	134
M2	Krylatskoye Rowing Canal	55.766868, 37.442611	0.18	3.0	Yes	24.8	8.89	148
M4	Moskva River, Krylatskoye district	55.753740, 37.448831	Length—473 km, basin area—17,600 km^2^, water discharge—109 m^3^/s		Yes	22.1	7.78	165
M6	Bolshoy Sadovy Pond	55.834221, 37.539308	0.082	2.0	Yes	23.6	8.77	103
M7	Nizhny Fermsky Pond	55.835205, 37.559273	0.067	1.8	Yes	21.5	8.93	100
M9	Unnamed pond in Odintsovsky district	55.765066, 37.446576	0.015	1.0	No	25.1	8.50	154
M10	Meshchersky Pond	55.674680, 37.410679	0.103	2.2	Yes	28.1	9.50	111
M13	Unnamed pond in Marfino district #1	55.689277, 37.365546	0.020	1.1	No	27.6	9.02	103
M14	Unnamed pond in Marfino district #2	55.668461, 37.385979	0.025	1.0	No	27.1	8.51	152
M16	Pervy Kamensky Pond	55.833901, 37.608266	0.055	1.5	Yes	28.1	8.41	172
M18	Patriarshiy Pond	55.763396, 37.592456	0.021	1.2	Yes	25.5	9.37	95
M19	Clean Pond	55.761603, 37.644501	0.038	1.4	Yes	25.6	8.81	131

**Table 4 toxins-17-00506-t004:** Primers used to amplify cyanotoxins biosynthesis genes.

Gene	Primer	T °C Annealing	Sequence (5′–3′)	References
*mcy*E, *nda*F	HEPF	52	TTTGGGGTTAACTTTTTTGGGCATAGTC	[46]
	HEPR		AATTCTTGAGGCTGTAAATCGGGTTT	
*mcy*A	mcyACdF;	59	AAAAGTGTTTTATTAGCGGCTCAT	[47]
	mcyACdR		AAAATTAAAAGCCGTATCAAA	
*mcy*B	McyB-F;	55	AGACCAAAAATTAACCTATCAACAG	[48]
	McyB-R		TACTAATCCCTATCTAAACAC	
*mcy*E	mcyE-F2	56	GAAATTTGTGTAGAAGGTGC;	[49]
	mcyE-R4		AATTCTAAAGCCCAAAGACG	
*ana*C	anaC-genF	58	TCTGGTATTCAGTCCCCTCTAT	[50]
	anaC-genR		CCCAATAGCCTGTCATCAA	
*cyr*B	cyrB M13f	58	GGCAAATTGTGATAGCCACGAGC	[51]
	cyrB M14r;		GATGGAACATCGCTCACTGGTG;	

## Data Availability

Strains analyzed herein are housed at Culture and Barcode Collection of Microalgae and Cyanobacteria “Algabank” (CBMC) in K. A. Timiryazev Institute of Plant Physiology RAS, Moscow, Russia. The sequences obtained during the current study are available in the NCBI SRA database (GenBank). Additional data requests should be addressed to E. Kezlya.

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
