# Peer review of "Comprehensive Study of Some Cyanobacteria in Moscow Waterbodies (Russia), Including Characteristics of the Toxigenic Microcystis aeruginosa Strains"

_toxins, 2025, doi:10.3390/toxins17100506_

Round 1
Reviewer 1 Report
Comments and Suggestions for Authors
This manuscript presents a valuable polyphasic study of cyanobacteria from Moscow water bodies. The work is comprehensive, but there are three major concerns regarding the molecular analysis that need to be addressed to strengthen the conclusions.
-
The phylogenetic trees are presented, but they are not sufficient on their own for accurate taxonomic assignment. The authors should calculate and report the 16S rRNA gene sequence similarity percentages between their novel strains and the most closely related, validly described species (especially the type species) from GenBank. This information is crucial for supporting the species-level identifications.
-
The authors state that the 16S rRNA gene amplicons were sequenced directly after PCR, not cloned. To validate the accuracy of the sequences, which forms the basis of the phylogenetic analysis, it is essential to demonstrate that the sequences are clean and unambiguous. Please provide the sequencing chromatograms (trace files) for the 16S rRNA gene sequences as supplementary material to confirm the absence of mixed peaks or sequencing errors.
-
For a robust phylogenetic analysis, the inclusion of type species for the respective genera is standard and necessary. The phylogenetic trees in Figure 7 and Figure 8 should be reconstructed to include the 16S rRNA gene sequences from the type species of the major genera discussed (e.g., Microcystis, Woronichinia, Dolichospermum, Anabaena, Argonema). This will provide a solid framework for the phylogenetic placement of the newly isolated strains.
Author Response
Dear Reviewer,
Thank you very much for your important comments!
The authors provide answers to your comments by using text in bold.
- The phylogenetic trees are presented, but they are not sufficient on their own for accurate taxonomic assignment. The authors should calculate and report the 16S rRNA gene sequence similarity percentages between their novel strains and the most closely related, validly described species (especially the type species) from GenBank. This information is crucial for supporting the species-level identifications.
Response. Corrected. We added the estimate the degree of similarity of the 16S rRNA gene sequence between gene sequences of type strains and own strains species-level identificated (Argonema galeatum, Woronichinia naegeliana and Microcystis aeruginosa). Supplementary tables S1-S3. Also we added type strains of Anabaena, Dolichospermum and Aphanizomenon in to the phylogenetic tree.
We added the description of this results in to “Results”, part “2.2. Molecular analysis”.
- The authors state that the 16S rRNA gene amplicons were sequenced directly after PCR, not cloned. To validate the accuracy of the sequences, which forms the basis of the phylogenetic analysis, it is essential to demonstrate that the sequences are clean and unambiguous. Please provide the sequencing chromatograms (trace files) for the 16S rRNA gene sequences as supplementary material to confirm the absence of mixed peaks or sequencing errors.
Response. Corrected. We added the sequencing chromatograms of the 16S rRNA gene sequences to Supplementary Material, File S10
- For a robust phylogenetic analysis, the inclusion of type species for the respective genera is standard and necessary. The phylogenetic trees in Figure 7 and Figure 8 should be reconstructed to include the 16S rRNA gene sequences from the type species of the major genera discussed (e.g., Microcystis, Woronichinia, Dolichospermum, Anabaena, Argonema). This will provide a solid framework for the phylogenetic placement of the newly isolated strains.
Response. For Woronichinia naegeliana the sequence from the type strain (OLE35S01) presents in the tree. For Argonema galeatum the sequences from the type strains (A003/A1, A003/B4, A003/B5, A003/D3, A003/D1) present too.
We corrected trees in Figure 7 and Figure 8 and added type strains for Microcystis aeruginosa (NIES 104, 1082, 102, 1123 and HUW226), Dolichospermum flos-aquae (1tu30s4, 04-10, AWQC112D, PCC9302, NIES-75), Aphanizomenon flos-aquae (CCAP 14017, 1tu29s19, PCC7905), Anabaena oscillarioides (BECID22, BECID32).
The choose of type strains was based on “Herdman, M. & Rippka, R. (2024). CYANOBACTERIOTA: Phylogeny and Taxonomy. cyanophylogeny.scienceontheweb.net.”
Reviewer 2 Report
Comments and Suggestions for Authors
The authors present a work aiming to test the urban waters of the Moscow megalopolis for the presence of toxigenic cyanobacteria. No doubt, that such work is necessary to conduct and present, both for the public, scientific community and the water management, health and environmental authorities. Overall, the paper was easy to read and follow, however i have some remarks and suggestions for further improvement.
- The title: I would remove "comprehensive", since except for two microcystis strains, all other isolates were only described at the level of morphology and16S sequences, followed by phylogeny reconstruction. In order to keep "comprehensive", i would suggest to include HPLC and MS profiles of all other strains, with suggested product assignments.
- Introduction is good as is
- Results: 16S sequences of newly isolated strains have to be submitted to the GenBank, possibly with delayed release, but the accession numbers have to be provided. It is a common practice.
- HPLC-HRMS. Here i have the following remarks: The precise identification of teh MC variants is not convincing. Most of the MC listed have known, and probably unknown isomers. I checked a few masses with CyanoMetDB, some of them correspond to 3-4 different MCs. The precise identification ideally would be fragmentation patterns, compared to databases, literature and standards. Authors mention the standards, but i did not see them being used in the paper. One also could show HPLC runs with standards for comparison. In any case, my major objective here- masses might be assigned to wrong molecules. Provide harder evidence, please.
- In order to keep word "comprehensive" in the title i would suggest to add HPLC- MS profiles of the other strains from this study. In the discussion, authors mention known Woronichinia products. Also a long list of compounds produced by Oscilatoriales, Anabaena, Aphanizomenon exists. Comparison of the profiles with published data would add more substance to the work.
- PCR screening: authors have to mention that the method alone does not prove 100% the presence of the corresponding gene clusters, since NRPS/PKS gene clusters are modular, and related modules may appear in different operons coding synthesis of completely different product. In this study Mcy is confirmed by presence of microcystins. So this is not an issue, but rarther a note.
- Methods: Please specify settings for ultrasonic bath.
- What sequencing kit did you use and what were the conditions.
Author Response
Dear Reviewer,
Thank you very much for your important comments!
The authors provide answers to your comments by using text in bold.
- The title: I would remove "comprehensive", since except for two microcystis strains, all other isolates were only described at the level of morphology and16S sequences, followed by phylogeny reconstruction. In order to keep "comprehensive", i would suggest to include HPLC and MS profiles of all other strains, with suggested product assignments.
Response. In our opinion, we can use "comprehensive" in the title, since in addition to morphology and phylogeny, all strains were screened for the presence of gene clusters encoding cyanotoxins by PCR. After initial screening by PCR, two strains were selected for confirmation of microcystin content by HPLC.
- Introduction is good as is
Response. Thank you for your positive comment.
- Results: 16S sequences of newly isolated strains have to be submitted to the GenBank, possibly with delayed release, but the accession numbers have to be provided. It is a common practice.
Response. Corrected. We submitted 16S sequences of newly isolated strains to the GenBank. Accessions have already been provided for the following sequences:
Anabaena sp. CBMC473m PX326131
Aphanizomenon sp. CBMC479m PX326132
Argonema galeatum CBMC475m PX326133
Dolichospermum sp. CBMC469m PX326134
Microcystis aeruginosa CBMC523m PX326135
Microcystis aeruginosa CBMC403m PX326136
They are included in supplementary files S4, S7. The remaining sequences of Woronichinia naegeliana are already submitted to GenBank. We are waiting for the accessions to be assigned soon.
- HPLC-HRMS. Here i have the following remarks: The precise identification of teh MC variants is not convincing. Most of the MC listed have known, and probably unknown isomers. I checked a few masses with CyanoMetDB, some of them correspond to 3-4 different MCs. The precise identification ideally would be fragmentation patterns, compared to databases, literature and standards. Authors mention the standards, but i did not see them being used in the paper. One also could show HPLC runs with standards for comparison. In any case, my major objective here- masses might be assigned to wrong molecules. Provide harder evidence, please.
Response. We thank the reviewer for the valuable comments. We have added missing information confirming our identification. Fragment spectra of standards and identified MC variants are presented in the Supplementary materials File S3, Figures S5-S8). Explanation of the identification procedure and characteristic ion has been added to the Results section.
This paragraph is given below:
“Identification was based on the exact masses of MCs and the data from the fragment spectra. Fragment spectra obtained for the identified compounds and standards corresponding to each other (Supplementary materials File S3, Figures S5-S8). All of the identified compounds belong to the microcystin group because of presence some of m/z corresponding to characteristic fragments of microcystin moleculars, such as 135,08 (Adda-fragment [C9H11O]+), m/z 213,09 ([Glu-Mdha+H]+), m/z 375,19 ([C11H15O+Glu+Mdha]+) in the fragment spectra. Other characteristic m/z values for each structural variant of MC and corresponding structure fragments are listed in the Table 2.
Table 2 Characteristic m/z values for each structural variant of MC and corresponding structure fragments”
- In order to keep word "comprehensive" in the title i would suggest to add HPLC- MS profiles of the other strains from this study. In the discussion, authors mention known Woronichinia products. Also a long list of compounds produced by Oscilatoriales, Anabaena, Aphanizomenon exists. Comparison of the profiles with published data would add more substance to the work.
Response. Thank you for your suggestion. However, we did not set the goal of studying the spectrum of compounds produced by non-toxigenic species. This study is aimed solely at finding toxin-producing strains in order to study their distribution, development under anthropogenic load on water bodies and to draw the attention of city authorities to the need to monitor blooms, especially in recreational water bodies.
- PCR screening: authors have to mention that the method alone does not prove 100% the presence of the corresponding gene clusters, since NRPS/PKS gene clusters are modular, and related modules may appear in different operons coding synthesis of completely different product. In this study Mcy is confirmed by presence of microcystins. So this is not an issue, but rarther a note.
Response. Thank you, we have added this note to the section 3 “Discussion”, line 285.
- Methods: Please specify settings for ultrasonic bath.
Response. Thank you, we have added settings for ultrasonic bath:
FinnSonic OY M03, Finland. US frequency 40 kHz, US power 80W/ 160W (nominal/ peak). Treatment time (extraction) - 20 minutes).
- What sequencing kit did you use and what were the conditions.
Response. Corrected. We added next sentence:
16S rRNA fragments of gene were sequenced on both strands using forward and reverse PCR primers and the Big Dye system (Applied Biosystems, USA), followed by electrophoresis using a Genetic Analyzer 3500 sequencer (Applied Biosystems, Waltham, MA, USA).
Round 2
Reviewer 2 Report
Comments and Suggestions for Authors
All my concerns and remarks has been taken care of. The paper is good for publication.